# Variation of the Tegmen and Cercus in *Sinopodisma rostellocerca* (Orthoptera: Acrididae: Melanoplinae) with Proposal of a New Synonym

**DOI:** 10.3390/insects15070526

**Published:** 2024-07-12

**Authors:** Renjie Qiu, Yuchen Yan, Hanqiang Wang, Jianhua Huang

**Affiliations:** 1Key Laboratory of Forest Bio-Resources and Integrated Pest Management for Higher Education in Hunan Province, Central South University of Forestry and Technology, Changsha 410004, China; 20211100012@csuft.edu.cn (R.Q.); yanyc@csuft.edu.cn (Y.Y.); 2Hunan Provincial Key Laboratory for Control of Forest Diseases and Pests, Central South University of Forestry and Technology, Changsha 410004, China; 3Shanghai Entomological Museum, Chinese Academy of Sciences, Shanghai 200032, China; whq@cemps.ac.cn

**Keywords:** intraspecific variation, *Sinopodisma rostellocerca*, *Sinopodisma hengshanica*, new synonym

## Abstract

**Simple Summary:**

The intraspecific variation of the tegmen and cercus in *Sinopodisma rostellocerca* was examined, the variation patterns were summarized, and the relationship between *S. rostellocerca* and *S. hengshanica* was discussed. The results showed that all types of variation in *S. hengshanica* fell into the range of variation in *S. rostellocerca*, leading to the disappearance of the boundary between the two species. Therefore, *S. hengshanica* was herein considered as a new junior synonym of *S. rostellocerca*.

**Abstract:**

Intraspecific variation is ubiquitous from individual traits to population level and plays an important role in a variety of fields. However, it is often ignored by systematists and comparative evolutionary biologists. In view of the limited knowledge of intraspecific variation, morphology-based identification has hindered the recognition of species borders and led to a great number of problems in the field of taxonomy and systematics. In this study, the intraspecific variation of the tegmen and cercus in *Sinopodisma rostellocerca* was examined, the variation patterns were summarized and the relationship between *S. rostellocerca* and *S. hengshanica* was discussed. The results showed that the intraspecific variation in the tegmen and male cercus was mainly manifested in the length and shape of the apical margin and dorso- and ventro-apical angles; this substantial variation occurred not only among intrapopulation individuals but also between the different sides of the same individuals, and all types of variation in *S. hengshanica* fell into the range of variation in *S. rostellocerca*, leading to the disappearance of the boundary between the two species. Therefore, *S. hengshanica* was herein considered as a new junior synonym of *S. rostellocerca*.

## 1. Introduction

The genus *Sinopodisma* Chang, 1940 was established with *Indopodisma pieli* Chang, 1940 as the type species [1]. There are 44 known species in the genus worldwide so far [2]. Except for *S. aurata* Ito, 1999 [3] and *S. punctata* Mistshenko, 1954 [4], distributed in Japan, all remaining species are endemic to China. *Sinopodisma* is a micropterous genus and belongs to the grasshopper subfamily Melanoplinae, among which some groups show some a degree of variation in the tegmen. For example, the genus *Podisma* contains both micropterous and apterous species, and there is some degree of reduction in the tegmen in the genus *Parapodisma,* varying from brachypterous through subbrachypterous and micropterous to subapterous [5].

*S. rostellocerca* You, 1980 was described based on the types from the Huaping Nature Reserve, Longsheng County, Gaungxi, south China, and is distributed in the northern region of Guangxi and from the southwestern region to parts of the northwestern region of Hunan [6,7,8,9,10]. The most distinct diagnostic character for *S. rostellocerca* is the apically rostriform cercus in males. There are six species in *Sinopodisma* with a rostriform or similar apex of the cercus in males, but *S. wuyishana* Zheng, Lian and Xi, 1985 [11] and *S. rufofemoralis* Fu and Zheng, 2004 [12] can be easily distinguished from *S. rostellocerca* by their reddish hind femora, and *S. tsaii* (Chang, 1940) [1] has a smaller body size and a much more yellowish body color in its living condition than *S. rostellocerca*. The species most similar to and difficult to separate from *S. rostellocerca* are *S. spinocerca* Zheng and Liang, 1986 [13] and *S. hengshanica* Fu, 1998 [14]. The minor difference among these three species is the apical shape of the cercus in males according to original descriptions, i.e., the ventro-apical denticle of the apex of the cercus in the male *S. hengshanica* is located at the ventral margin but not at the apex as in the male *S. spinocerca*, and the apex of the cercus in the male *S. hengshanica* is rounded with a denticulate but not rostriform ventro-apical angle as in the male *S. rostellocerca* [13,14]. However, variation occurs not only among species just like in the genera *Podisma* and *Parapodisma* mentioned above, but also among individuals within the same species. For example, distinct intraspecific variation in the shape of the tegmen and male cercus were observed frequently in *S. rostellocerca* during our investigations in the field, making the boundaries among these three species more ambiguous.

Intraspecific variation or polymorphism is the variation found among individuals within a species and is defined as being at least partly independent of ontogeny and sex [15]. In fact, intraspecific variation is ubiquitous from individual traits (physiological, morphological, phenological, ethological, etc.) to population level (vital rate, etc.), is the major focus of research on microevolution, and plays an important role in a variety of fields such as population genetics, systematics, comparative evolutionary biology, phylogeography, and so on [15,16].

However, polymorphism is often ignored by those who study macroevolution: systematists and comparative evolutionary biologists [15]. Despite the prevalence of intraspecific variation, phylogenetic biologists have a long and continuing tradition of ignoring polymorphism. For example, morphological systematists often exclude characters that show any or “too much” variation within species. Both molecular and morphological systematists often “avoid” or minimize polymorphism by sampling only a single individual per species. In view of the limited knowledge of intraspecific variation, morphology-based identification has hindered the recognition of species borders and affected the views on the actual distribution of some species [17]. Considerable intraspecific variation has led to the descriptions of a large number of nominal species, misidentification, and a great number of inconsistently synonymized species in some groups of organisms such as Mollusca [18].

As a group of bilaterally symmetrical organisms, insects have many binate internal and external organs which are usually equivalent to (or mirror images of) each other in shape and structure. However, intraspecific variation exists not only among different individuals but also between the different sides of the same individual sometimes, resulting in an asymmetry between the left and right components of the binate organ. In this study, we examined a great number of individuals of *S. rostellocerca* from Hunan and Guangxi and found substantial variation in the tegmen and male cercus not only among individuals of the same population but even between the different sides of the same individuals. A similar pattern of variation was also found in *S. hengshanica,* and all the so-called distinguishing characters of *S. hengshanica* can be involved in the range of variation in *S. rostellocerca*. In this paper, we present evidence that *S. hengshanica* should be regarded as a new junior synonym of *S. rostellocerca*.

## 2. Material and Methods

This paper is based on specimens of *S. rostellocerca* and all of the types of *S. hengshanica* deposited at the insect collections of the Central South University of Forestry and Technology (CSUFT), China as well as paratypes of *S. rostellocerca* deposited at the Shanghai Entomological Museum, Chinese Academy of Sciences, China (SEMCAS). All photographs were taken using a Nikon D600 digital camera (Nikon Inc., Tokyo, Japan) or Leica DFC 5500 system (Leica Microsystems Inc., Weizler Hessen, Gamany), and the images were stacked using Helicon Focus version 6.0. The terminology for morphology follows Uvarov (1966) [19] and Storozhenko et al. (2015) [20]. The terminology for describing the shape of the cercus is paraphrased briefly as follows:

Concave: curving inward.

Conical: relating to or resembling a cone.

Cylindrical: related to or having the form/shape or properties of a cylinder or tube.

Rostriform: beaklike, i.e., resembling the beak of a bird in shape.

Rounded: curving and somewhat round in shape rather than jagged.

Spear-shaped: relating to or resembling a spear in shape.

Truncate: terminating abruptly by having or as if having an end or point cut off.

Frustum-cone-like: relating to or resembling a truncated cone or pyramid in shape (frustum is the part that is left when a cone or pyramid is cut by a plane parallel to the base and the apical part is removed).

A total of 328 individuals were examined, including 134 males and 194 females (Table 1). The individuals examined were composed of the holotype male, two paratye males, and four paratype females of *S. hengshanica* as well as of four paratype males, one paratype female, 127 common males, and 189 common females of *S. rostellocerca* from 9 localities. A total of 31 males and 33 females of *S. rostellocerca* were stochastically selected for careful comparison, of which 20 males and 19 females were photographed. The four base measurements generally used for grasshoppers, i.e., body length, pronotum length, tegmen length, and hind femur length of the 328 examined individuals were gauged using a digital vernier caliper or an ocular micrometer in a stereomicroscope. The mean and standard deviation of the measurements were computed using functions “mean()” and “sd()” in the “base” and “stats” packages of R 4.4.0, respectively. Variance analysis was implemented using the function “aov()” in the “stats” package. Box-and-whisker plots were produced using the function “boxplot()” in the “graphics” package. Correlation analysis between the variation of the male cercus and tegmen was carried out based on the data from the 20 males of *S. rostellocerca* and the three types of males of *S. hengshanica* which were photographed and enabled us to compare carefully and repeatedly. The quantitative matrix for correlation analysis was generated according to the coding criteria described below and then processed using function”corrplot()” in the “corrplot” package of R 4.4.0. To test the reliability of the relative length of the tegmen to a specified point of reference (or the correlation between relative and absolute length) within the species, both the relative and absolute length of the tegmen were involved in the data matrix for correlation analysis.

The coding criteria of the characters are as follows:

(1) The criteria for coding shape of cercus,

Apical margin of the cercus: concave: 1; straight and vertical: 2; straight and inclined forwards: 3; straight and inclined backward: 4; deeply notched: 5; rounded: 6.

The dorso-apical angle of the cercus: rectangularly rounded: 1; broadly rounded: 2.

The ventro-apical angle of the cercus: rounded: 1; angulate: 2; denticulate: 3 dentate: 4; spinous: 5.

The whole shape of the apex of the cercus: rostriform: 1; conical: 2; spear-shaped: 3 truncate: 4; rounded: 5; roundly truncate: 6.

(2) The criteria for coding relative length of tegmen,

Just reaching the posterior margin of the metathorax: 0;

Just reaching half of the first abdominal tergite: 1;

Just reaching two thirds of the first abdominal tergite: 2;

Just reaching three fourths of the first abdominal tergite: 3

Distinctly not reaching the posterior margin of the first abdominal tergite: 4;

Slightly not reaching the posterior margin of the first abdominal tergite: 5;

Just reaching the posterior margin of the first abdominal tergite: 6;

Slightly exceeding the posterior margin of the first abdominal tergite: 7;

Distinctly exceeding the posterior margin of the first abdominal tergite: 8.

The institutional acronyms are as follows:

CSUFT: Insect collection of the Central South University of Forestry and Technology, Changsha, China (curator: Jianhua Huang).

SEMCAS: the Shanghai Entomological Museum, Chinese Academy of Sciences, Shanghai, China (curator: Haisheng Yin).

## 3. Results

### 3.1. Variation of Tegmen

#### 3.1.1. Variation of Tegmen in *S. rostellocerca*

When a great number of individuals were examined, much variation was found between the left and right tegmina in males of *S. rostellocerca*. The asymmetry is mainly manifested in the length and the shape of the dorso- and ventro-apical angles as well as of the apical margin. As seen in Figure 1a,b, the left tegmen is normally oval in shape, but the right one obviously reduced and is dysmorphic with the apical margin bifurcate, with the lower fork forming a digitation appearance. In Figure 1c,d, the left tegmen is distinctly smaller than the right one with the apical margin straight and vertically truncate. In Figure 1e,f, the apical margin of the left tegmen is straight and obliquely truncate, and the ventro-apical angle is angulate, but the apical margin and ventro-apical angle of the right tegmen are broadly rounded. In Figure 1g,h, the dorso-apical angle of the left tegmen is narrow and bluntly rounded, and the ventro-apical angle is bluntly angulate, but the dorso-and ventro-apical angles of the right tegmen are broadly rounded. Similar minor variation between the left and right tegmina of males of *S. rostellocerca* can be observed frequently in other individuals. Among the carefully compared males of *S. rostellocerca*, two individuals are distinctly asymmetrical and 16 individuals exhibit slight asymmetry in the shape and length of tegmen, representing 6.45% and 51.61% of the total, respectively. The remaining 13 individuals are symmetrical or nearly symmetrical in the shape and length of the tegmen, representing 41.94% of the total.

The asymmetry exists not only in males but also in females. Among the female specimens examined, the tegmen exhibits similar asymmetrical variation. As shown in Figure 2a–d, the tegmen may be normal in size at one side (Figure 2b,c) but strongly reduced at the other side (Figure 2a,d), with the most extreme situation of only reaching the posterior margin of the metathoracic tergite (Figure 2d). In the individuals with paired normal tegmina, the asymmetry will still display to some extent in the length of tegmen, the shape of dorso- and ventro-apical angles, and the apical margin. For example, in Figure 2e,f, the ventro-apical angle of the left tegmen is broadly rounded and the ventral margin is slightly straight, but the ventro-apical angle of the right tegmen is angulate and the ventral margin is distinctly convexed near the base. Among the carefully examined female individuals, three individuals are distinctly asymmetrical and 10 individuals exhibit slight asymmetry in the shape and length of the tegmen, representing 9.09% and 30.30% of the total, respectively. The remaining 20 individuals are symmetrical or nearly symmetrical in the shape and length of the tegmen, representing 60.61% of the total.

While asymmetry is observed in some individuals, the majority of individuals are symmetrical or approximately symmetrical in their paired organs. In this case, intraspecific variation emerges usually among conspecifics.

In males, the apical margin may be rounded (Figure 1a,h,i,l) or straight (Figure 1c,e,g,j,k), obliquely truncate (Figure 1g,j,k), or vertical (Figure 1c). The dorso-apical angle may be narrowly angulate (Figure 1e,g,h,j,k) or broadly rounded (Figure 1a,d,f,i,l). The ventro-apical angle may be bluntly angulate (Figure 1e,g,j) or broadly rounded (Figure 1a,d,f,h,i,k,l), situated close to (Figure 1a,c,d,i,j,l) or slightly distant from (Figure 1e,g,h,k) the apex. The curvature of the dorsal margin also varies from nearly straight (Figure 1c) to distinctly curved with the turning point situated near the apex (Figure 1a,d,e,l) or at the middle (Figure 1f–k) of the tegmen. The most conspicuous variation is in the length of the tegmen. The apex of the tegmen may reach only half or two-thirds of the first abdominal tergite (Figure 1c,d,i), or nearly reach (Figure 1a,j,k), or slightly or even distinctly exceed (Figure 1e–h,l) the posterior margin of the first abdominal tergite. The tegmen of males shows more diverse types of variation in length than in symmetry, with 3.23% just reaching half of the first abdominal tergite, 19.36% distinctly not reaching, 24.19% slightly not reaching, 12.90% just reaching, 16.13% slightly exceeding and 24.19% distinctly exceeding the posterior margin of the first abdominal tergite.

The female shows a similar pattern of variation in the tegmen, including the length and the shape (Figure 2g–l and Figure 3a–f). In addition, there are two individuals that have distinctly concave apical margins (Figure 3e,f) and one that has a nearly pointed dorso-apical angle (Figure 3f). The tegmen of females also has more diverse types of variation in length than in symmetry, with 3.03% just reaching the posterior margin of the metatergite, 7.57% just reaching half of the first abdominal tergite, 37.88% distinctly not reaching, 4.55% slightly not reaching, 15.15% just reaching, 15.15% slightly exceeding, and 16.67% distinctly exceeding the posterior margin of the first abdominal tergite.

#### 3.1.2. Variation of Tegmen in *S. hengshanica*

Although *S. hengshanica* has only a limited number of types available for examination and comparison, typical variation was still observed in both the male and female types. The apex of tegmen distinctly exceeds the posterior margin of the first abdominal tergite in one paratype male (Figure 3g), but just reaches it in another paratype male (Figure 3h), and does not reach it in the holotype male (Figure 3i). As for the four paratype females, the variation in tegmen length includes three kinds of situation, i.e., distinctly exceeding (Figure 3j), slightly not reaching (Figure 3k), and far distant from (Figure 3l) the posterior margin of the first abdominal tergite. In a word, *S. rostellocerca* and *S. hengshanica* have completely the same variation pattern in the tegmen, and all forms of tegmina in *S. hengshanica* can also occur in *S. rostellocerca.*

### 3.2. Variation of Cercus in Male

#### 3.2.1. Variation of Cercus in Male of *S. rostellocerca*

It seems that variation in the male cerci is more extensive than in the tegmen. Although the shape of the cercus in most individuals is typically rostriform apically, it indeed shows a completely different appearance sometimes (Figure 4b,c,f). If it does not occur asymmetrically in the same individuals along with a rostriform cercus, no one will be able to determine it as a form of variation. In Figure 4b, the right cercus is frustum-cone-like; in Figure 4c,f, the left and right cerci are spear-shaped. In Figure 4g, the left cercus is cylindrical with a rounded apical margin and a short and bluntly denticulate ventro-apical angle. However, the opposite cerci of these four individuals are all rostriform (Figure 4a,d,e,h). In the individuals with a paired rostriform cerci, there is still a distinct difference between the left and right cerci. In Figure 4i,j, the apical margin of the left cercus is straight, bisinuate, and vertically truncate (Figure 4i), but that of the right one is evenly rounded (Figure 4j). In Figure 4k,l, the apical margin of the left cercus is straight and posteriorly obliquely truncate (Figure 4k) but that of the right one is distinctly concave in the middle (Figure 4l).

Besides the variation within the same individual, the variation among different individuals mainly include the shape of the apical and ventral margins as well as the dorso- and ventro-apical angles. The apical margins may be concave (Figure 4l,m,o–q), straight (Figure 4a,d–f,h,i,k,r–x), or rounded (Figure 4g,j), vertically (Figure 4d–f,h,i,r–u), or obliquely truncate forwards (Figure 4n,v), or backward (Figure 4a,k,w,x). The dorso-apical angle may be roundly rectangular (most of the time such as in Figure 4d,e,h,i) or broadly rounded (Figure 4a,k,x). The ventro-apical angle may be long and spinous (Figure 4r), robustly dentate (Figure 4q), finely denticulate (Figure 4i,j,n,s), or angulate (Figure 4g,t,u). The ventral margin is usually straight but sometime may be slightly sinuate (Figure 4a,d,p,r,x).

#### 3.2.2. Variation of Cercus in Males of *S. hengshanica*

Similar to the tegmen, the variation pattern of the male cercus of *S. hengshanica* also completely agrees with that of *S. rostellocerca*. The left and right cerci of the holotype male of *S. hengshanica* are a little asymmetrical. The apical margin of the left cercus is straight and slightly obliquely truncates forwards (Figure 4y), and that of the right cercus is rounded (Figure 4z). The ventro-apical angle is angulate at the left (Figure 4y) but denticulate at the right cercus (Figure 4z). One of the paratype males has the most typical rostriform cercus with a concave apical margin and a long spinous ventro-apical angle (Figure 4aa), but the shape of the cercus of another paratype male is a little different with a straight and vertical apical margin straight, and a sharply dentate ventro-apical angle (Figure 4bb). All forms of cerci in *S. hengshanica* also can be found in *S. rostellocerca*.

### 3.3. Correlation Analysis between Tegmen and Cercus in Males

To explore the correlation between the variation of the tegmen and cercus of male *S. restellocerca*, the overall apical shape, apical margin, dorso-, and ventro-apical angles of the male cercus as well as the relative and absolute length of the tegmen were coded into a numerical value to generate a quantitative matrix for correlation analysis (Table 2). The result shows that there is nearly no significant correlation between the variation of the cercus and tegmen in males (Figure 5). As seen in Figure 5, the majority of character pairs are uncorrelated. The significant correlation between the overall apical shape and the ventro-apical angle of the cercus indicates that a ventro-apical angle is a major factor that determines the overall shape of the apex of the cercus in males. Considering the majority of pairs of characters are uncorrelated, the significant correlation between a few pairs of characters in the male cercus and tegmen may represent no biological implication. However, the strong correlation between the relative and absolute length of the tegmen confirms well that the relative length of the tegmen is reliable within species for comparing tegmen size when used for morphological description.

### 3.4. Taxonomy [2,3,21,22,23,24,25,26,27,28,29,30,31,32,33,34,35]

#### 3.4.1. Genus *Sinopodisma* Chang, 1940

*Indopodisma* (*Sinopodisma*) Chang, 1940: 40, 68.

*Sinopodisma* Chang; Bey-Bienko and Mistshenko, 1951: 239; Mistshenko, 1952: 446; Rehn and Randell, 1963: 10; Zheng, 1985: 164; Storozhenko, 1994[1993]: 19; Zheng, 1993: 121; Otte, 1995: 441; Yin, Shi and Yin, 1996: 643; Jiang and Zheng, 1998: 127; Ito, 1999: 503; Xu, Takeda and Zheng, 2003: 335–342; Xu, Zheng and Li, 2003: 99–105; Wang, Li & Yin, 2004: 100; Li and Xia, 2006: 300; Ito, 2015: 78; Grzywacz and Tatsuta, 2017: 18; Qiu et al., 2020: 23–42.

Type species: *Indopodisma pieli* Chang, 1940, by original designation.

Diagnosis. Body medium- to small-sized. Head shorter than pronotum. Face slightly reclined in profile, frontal ridge with median longitudinal sulcus and lateral margins nearly parallel. Vertex a little narrow, with interocular distance equal to or slightly less than width of frontal ridge between antennal sockets. Antennae filiform and narrow, exceeding the posterior margin of pronotum. Eyes large and oval. Pronotum cylindrical with fine punctation and rugulae; posterior margin circular arced with small triangular notch in middle; median keel low and lateral keel absent; lower posterior angles of the lateral lobes usually subangulate. Prosternal process conical with apex pointed. Tegmina reduced, squamiform, oval to lanceus, and laterally situated, not reaching or exceeding posterior margin of first abdominal tergite; anterior margin nearly straight or slightly arced. Hind femora with upper median keels smooth and spined apically or not, lower lateral genicular lobes rounded apically. Hind tibiae without ectoapical spine. Tympanum developed. Supra-anal plate triangular in males, with median longitudinal sulcus and short oblique carina each at both sides near the base; cerci cylindrical to conical, broad basally and slightly incurved, the shape of apex usually varies among different species; subgenital plate short conical and pointed apically. Ovipositor valves elongate, with upper valves denticulate at outer margins; subgenital plate triangularly protruded at middle of posterior margin.

Distribtion. CHINA, JAPAN.

Species composition. Currently there are 44 species listed in the genus *Sinopodisma* in the Orthoptera Species File [2].

#### 3.4.2. Species *Sinopodisma rostellocerca* You, 1980 [2,6,8,14,24,25,33,34,36]

(Figure 6a–v)

*Sinopodisma rostellocerca* You, 1980: 233; Zheng, 1985: 167; Wei and Yin, 1986: 297; Zheng, 1993: 125; Yin, Shi and Yin, 1996: 643; Jiang and Zheng, 1998: 129–130; Li and Xia, 2006: 314.

*Sinopodisma hengshanica* Fu, 1998: 188, 190; Li and Xia, 2006: 311; Li et al., 2022: 616. **New synonym.**

Type locality: China (Longsheng, Guangxi).

Male. Body medium- to small-sized. Head short and broad, half as long as length of pronotum. Face slightly oblique in profile view; frontal ridge longitudinally sulcate from below base of antennae to lower margin, with lateral margins nearly parallel and slightly constricted only below median ocellus. Vertex slightly inclined forwards, with fastigium slightly depressed in middle, interocular distance slightly narrow, approximately as broad as frontal ridge between antennal sockets. Eyes oval with vertical diameter 1.5 times horizontal diameter and 1.6 times as long as subocular furrow. Antennae filiform, extending beyond posterior margin of pronotum, with median segments about 2.5 times as long as broad. Pronotum cylindrical, with dorsum rugulose and finely punctate; anterior margin straight, posterior margin circularly arced with small triangular notch in middle; median carina low and distinct, lateral carina absent; three transverse sulci distinct with posterior sulcus located at postmedian part and distinctly interrupting median carina; prozona approximately two times as long as metazona; lateral lobes longer than high, with anteroventral angle rounded and posteroventral angle subrectangular. Prosternal process conical with apex pointed. Mesosternal lobes with interspace slightly broader than long and having the length about 1.2 times minimum width. Both tegmina and hind wings reduced; tegmen slender and narrow, oval and squamiform, about 2.2–3.0 times as long as broad, apex rounded or truncate, not reaching or just reaching or distinctly exceeding posterior margin of first abdominal tergite; hind wing long and narrow, substantially shorter than and completely covered by tegmen. Hind femora slightly slender, with upper carinae smooth and spined at apex; both upper and lower genicular lobes rounded. Hind tibiae with eight to nine spines at outer margin and eight to nine spines at inner margin, ectoapical spines absent. Abdomen with first three segments distinctly rugulose and punctate. Tympanum developed with oval aperture. Terminal (the tenth) abdominal tergite with pair of small circular furcula in middle of posterior margin. Supra-anal plate equilaterally triangular, with longitudinal sulcus and oblique carina each at both sides near base. Cerci cylindrical, slightly laterally compressed, directing dorsoposteriorly and reaching the apex of supra-anal plate; base broader than remaining part; apex usually rostriform with dorso-apical angle broadly or narrowly rounded, ventro-apical angle spinous, robustly angulate or denticulate, dorsal margin strongly concave in middle, apical margin concave, rounded or straight and vertical or inclined forwards or backward, and ventral margin straight or slightly sinuate. Subgenital plate short conical with apex pointed.

Epiphallus bridge-shaped, not divided into two symmetrical halves; bridge narrow, with anterior margin nearly straight and posterior margin strongly and broadly concave; lateral plate inverted triangular, with anterior projection thick and rounded, posterior projection narrowly and bluntly angulate, inner margin straight and oblique, outer margin slightly concave or nearly straight, and slightly inclined inwards, making whole outline of epiphallus a little trapezoid in dorsal view; lophi large and auriform in posterior view, projecting upwards, obliquely located along inner margin of lateral plate; ancorae well developed and conical, slightly projecting anterointeriorly with apex sharply or bluntly pointed and distinctly incurved ventrally. Phallic complex with valves of cingulum shorter than apical valves of penis; rami of cingulum having asmall vesicle at each side near apex of valves of cingulum; cingular apodeme narrow, as long as basal valves of pennis; zygoma narrow and circular arc (Figure 6o–v).

**Female**. Body larger and more robust than males. Eyes with vertical diameter as long as subocular furrow. Pronotum with prozona approximately 1.5 times as long as metazona. Cerci short conical. Upper valves of ovipositor broad and short with outer margins denticulate; lower valves with outer margins edentate except for big tooth at the base. Subgenital plate with posterior margin triangularly protruding in the middle. Other characters similar to males.

**Coloration**. Body yellowish brown. Postocular strip black, extending to anterior margin of tegmen. Tegmen blackish brown. Hind femur with upper and outer sides yellowish green, inner and lower sides yellow, knee black. Hind tibiae bluish green with base black and pale annulation near the base.

**Measurement**. Body length: male, 17.00–26.56 mm, female, 22.36–33.91 mm; pronotum length: male, 4.24–5.98 mm, female, 5.56–8.06 mm; tegmen length: 2.44–5.77 mm, female, 2.90–6.26 mm; hind femur length: male, 10.51–14.26 mm, female, 12.53–18.97 mm.

According to the original description [6,14] and Li et al. [25], the measurement of body and hind femur length of both males and females as well as that of the tegmen length of male *S. hengshanica* are slightly larger than make *S. rostellocerca,* but with only a tiny gap between them (Table 3). While the measurement of the pronotum length of male *S. hengshanica* is slightly larger than male *S. rostellocerca,* there is no gap between them, i.e., the minimum value for *S. hengshanica* is equivalent to the maximum value for *S. rostellocerca*. There is no significant difference in the measurement of the remaining characters between the two species (Table 3).

However, the value range of most measurements of both species are substantially expanded when a large number of individuals are measured (Table 4 and Table 5). When the paratypes of *S. hengshanica* were remeasured, we found that the original measurement of the body length of paratype females may be inaccurate. The value of body length of the four paratype females of *S. hengshanica* remeasured herein varies from 23.75 to 29.02 mm (Table 5), and no value reaches 31.00 mm. Meanwhile, the measurement of body length of females of *S. rostellocerca* from Huaping, the type locality, varies from 23.96 to 32.79 mm (Table 5), with its maximum value exceeding the maximum value of 31.50 mm recorded for the females of *S. hengshanica* in the original reference [14]. In addition, the measurement of the body length of males of *S. rostellocerca* in the population of Huaping, the type locality, varies from 17.00 to 23.72 mm (Table 5), with a mean of 21.29 ± 2.65 mm (Table 4) and its maximum falls into the range of that of *S. hengshanica* (Table 5). When the statistics were computed by populations, we found that there is no significant difference among most populations (Table 4 and Table 5; Figure 7). The body and tegmen length of males as well as the tegmen length of females in the Yaoshan population are significantly smaller than those in other populations (Figure 7), but is not representative statistically due to the single individual measured. The measurement of the Yangmingshan population is distinctly smaller on average than that of other populations most of the time, but the maximum of many characters falls into the range of other populations (Table 5, Figure 7). For example, the maximum of body length in males of the Yangmingshan population is 21.75 mm, distinctly exceeding the minimum of Xiaoxi, Gaowangjie, Jiemuxi, Maoershan, Gongcheng and Huaping populations (Table 5). The maximum body length in females of the Yangmingshan population is 30.58 mm, distinctly exceeding not only the minimum of all other populations but also the maximum of *S. hengshanica* which is 29.02 mm (Table 5). The maximum of other measurements of the Yangmingshan population in both male and female are all distinctly larger than the minimum of most of the other populations (Table 5).

The result of the variance analysis based on body length, pronotum length, tegmen length, and hind femur length shows that the difference is insignificant among most populations (Appendix A). For example, in the body length of males of the Yangmingshan population is significantly different from most of the other populations due to its distinctly smaller body size (Appendix A). However, it is not significantly different from the Huaping and Yaoshan populations, indicating that the body size of males in Huaping and Yaoshan is also slightly smaller than that in the other populations. Interestingly, the Huaping population is also not significantly different from the other populations in body length, demonstrating that the body size of the Huaping population is in the position between those of Yangmingshan and other populations. The Huaping population is just like a bridge filling the gap in body size between the Yangmingshan and other populations. 

**Material examined**. ***S. hengshanica*:** a holotype male, Hengshan, Hunan Province, CHINA, 30 August 1993, Peng Fu leg.; paratypes, one female, data the same as the holotype; two males and three females, Hengshan, Hunan Province, CHINA, August 1991, Peng Fu leg. (CSUFT). ***S. rostellocerca*:** paratypes, four males and one female, Baiyan, Huaping National Nature Reserve, Longsheng County, Guangxi, CHINA, 5 September 1962, collector unknown (SEMCAS, 14509968–14509972). In addition, more than three hundred individuals of *S. rostellocerc* from the type locality (Huaping, Longsheng county) and Maoershan, Xing’an County, Guangxi as well as southwest and west Hunan have been examined (Table 1).

**Distribution**. CHINA (Guangxi, Hunan).

**Remark**. According to the original description [14], *S. hengshanica* is most similar to *S. spinocerca*, and the main difference is the ventro-apical denticle of the cercus in males located at the ventral margin but not at the apex; it is also similar to *S. rostellocerca*, and the main difference is that the apex of the cercus in males is not rostriform but is rounded with a denticulate ventro-apical angle. However, the results of this study confirm that this distinguishing character of *S. hengshanica* is just a form of variation in *S. rostellocerca*. The holotype male of *S. hengshanica* has its cerci a little asymmetrical in an apical shape between the left and right sides (Figure 4y,z), and the two paratype males are significantly different from the holotype in the apical shape of the cercus which is typically rostriform and much more similar to that of *S. rostellocerca* (Figure 4aa,bb). The shape of the cercus represented by the right cercus of the holotype males of *S. hengshanica* (Figure 4z) was also observed in some individuals of *S. rostellocerca* (Figure 4j,n). There is no significant difference observed in other characters between *S. rostellocerca* and *S. hengshanica* such as in the measurement generally used in grasshoppers and male genitalia (Figure 6o–v). Since intraspecific variation in *S. rostellocerca* is very conspicuous not only among different individuals but sometimes even in the same individual, and all distinguishing characters for identifying *S. hengshanica* are involved among the range of variation in *S. rostellocerca*, we consider herein *S. hengshanica* a junior synonym of *S. rostellocerca.*

## 4. Discussion

### 4.1. Non-Neglectable Intraspecific Variation

New species are recognized and described most of the time based on only a few types or sometimes even a single type. For example, *S. rostellocerca* was described by You [3] based on one holotype male, four paratype males, and four paratype females, and the range of measurement was slightly narrow (Table 3). *S. hengshanica* was described by Fu based on one holotype male, one paratype male, and two paratype females [14]. However, the wide range of actual variation within species, i.e., intraspecific variation, may not be fully revealed under these circumstances. For example, the range of the measurement of *S. rostellocerca* was slightly expanded in Li et al. (Table 3) [25], and it was further expanded distinctly in this study when a great number of individuals were examined (Table 5; see the value of the Huaping population with pop_id “Sr_gxhp”).

Grasshoppers are extremely diverse in terms of size, body shape, feeding biology, ecology, and life-history traits [37,38,39]. This variation not only exist between species but also occur within the same species sometimes. For example, the shape of the species-specific male cerci in Melanoplinae has been found to vary within species in *S. rufofemoralis* Fu and Zheng, 2004 [12] and the shape of male genitalia was confirmed to be labile in the *Melanoplus scudderi* complex [40]. Wing polymorphism in European Orthoptera is especially common in Tetrigidae and Gryllidae [41]. Within Acrididae, macropterous specimens occur occasionally in several brachypterous species mainly in the genera *Chrysochraon*, *Euthystira*, *Chorthippus*, *Melanoplus,* and *Podisma*. Although *Podisma pedestris* is generally a micropterous flightless species, long-winged individuals (macropterous forms) have been frequently described in the literature [41,42]. Indeed, wing polymorphism is an adaptation at the root of their ecological success [43]. In this study, we also observed a conspicuous variation in the shape of male cercus and the length of tegmen in *S. restellocerca* which sometimes even occurred between the right and left components of the paired structures in the same individual. Furthermore, we also found in this study that the neglect of this intraspecific variation had already caused the description of a junior synonym, *S. hengshanica* [14]. Therefore, intraspecific variation should not be neglected at any time because sometimes even distinct and discrete (or qualitative) characters also do not represent independent species with certainty [44], and significant differences in morphology might not always perfectly reflect the correct natural classification of the related taxa [45].

### 4.2. Independent Species or Intraspecific Variation?

While new species are usually determined and described by morphological taxonomists mainly based on the significant differences of a few distinguishing characters, there is no agreed standard to date on what degree of difference is significant and represents an independent species. The main analytical task of species-level systematics is to distinguish between intraspecific and interspecific character variation, i.e., polymorphism and the fixed (or nearly fixed) diagnostic features. The delimitation, diagnosis, and description of species is at least as important an endeavor of systematics as phylogenetic reconstruction. While it is difficult to identify differences between interspecific and intraspecific variation [46], some methodological efforts may facilitate to some extent the resolution of this issue. Firstly, denser sampling and more critical morphological examination are needed to reveal a wider range of variation both between and within species [47,48,49]. In this study, the careful examination and comparison of much more material than before revealed conspicuous intraspecific variation in the tegmen and cercus of *S. restellocerca,* providing us with very strong evidence to clarify the relationship between *S. restellocerca* and *S. hengshanica*. Secondly, quantitative analysis using morphometric and geometric methods should be used to help distinguish between interspecific and intraspecific variation more reliably [46]. Recently, these approaches have been increasingly used to reveal in many groups of grasshoppers the variation of locomotory morphology, antenna length, head shape [50,51,52,53,54], mandibles [55], wings [50,54,56], and so on. In this study, the quantitative analysis of the measurement shows that there is no significant difference in body size between *S. restellocerca* and *S. hengshanica*. While the body size of the Yangmingshan population is slightly smaller than that of other populations, it does not necessarily represent an independent species because body size may be constrained by short growing and reproductive seasons in seasonal environments resulting from altitudinal variation or highly heterogeneous topography, and large body size is usually associated with places having a longer temporal window for development and reproduction [57]. Finally, with the generation of molecular data becoming easier and cheaper, the discrimination between intraspecific variation and interspecific variation has increasingly benefited from molecular methods [17,58]. While we are not able to use molecular data in this study to compare the intraspecific variation of *S. rostellocerca* with those of *S. hengshanica* due to the unavailability of molecular data of *S. hengshanica* (no additional material was collected from the type locality hitherto except the types), the genomic data are expected to be used to test the relationship of *S. rostellocerca* with another closely related congeneric species, *S. spinocerca*, in the near future. The true phylogenetic relationship will be revealed more perfectly only based on a framework of integrative taxonomy [45].

## Figures and Tables

**Figure 1 insects-15-00526-f001:**
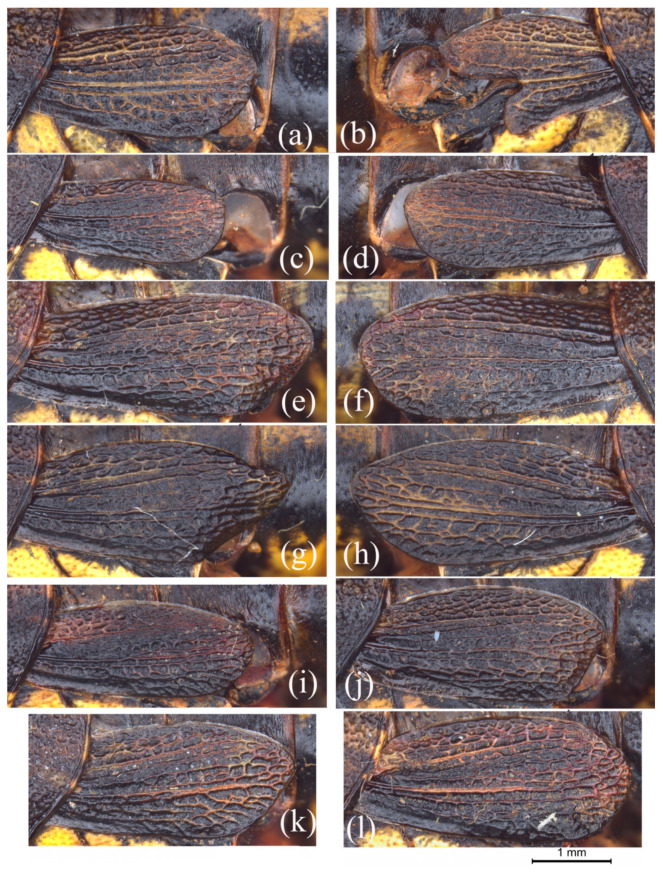
Variation of the tegmen in the males of *Sinopodisma rostellocerca*. (**a**–**h**) Tegmina of four male individuals of *S. rostellocerca* showing the asymmetrical variation between the left and right tegmina of the same individuals. Images (**a**–**h**) represent the left and right tegmina of the same individual, respectively. (**i**–**l**) Tegmina of the male individuals of *S. rostellocerca* showing the variation pattern of the tegmen among different individuals. Scale bar: 1 mm.

**Figure 2 insects-15-00526-f002:**
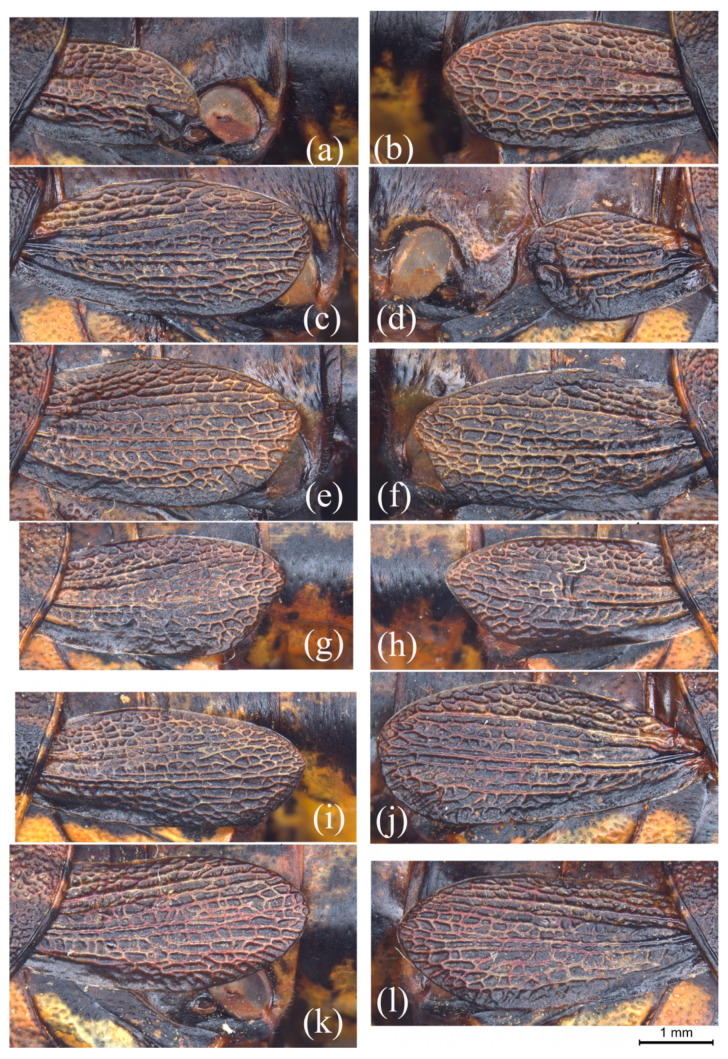
Variation of tegmen in the females of *Sinopodisma rostellocerca*. (**a**–**f**) Tegmina of three female individuals of *S. rostellocerca* showing the asymmetrical variation between the left and right tegmina of the same individuals. (**g**–**l**) Tegmina of the female individuals of *S. rostellocerca* showing the variation pattern of tegmen among different individuals. Images (**a**–**h**,**k**,**l**) represent the left and right tegmina of the same individual, respectively. Scale bar: 1 mm.

**Figure 3 insects-15-00526-f003:**
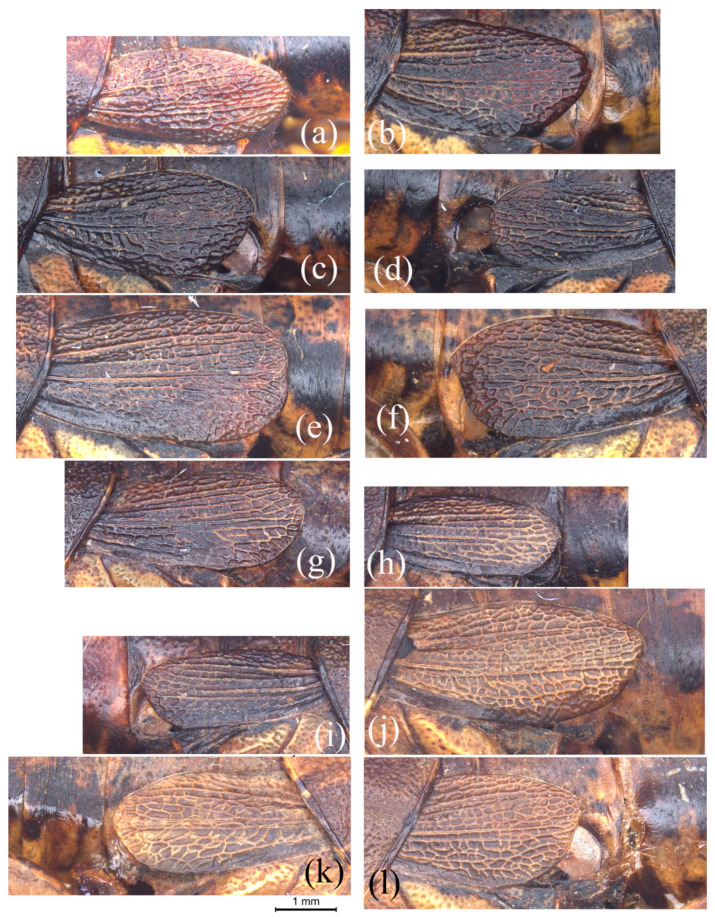
Variation of tegmen in the females of *Sinopodisma rostellocerca* and *S. hengshanica*. (**a**–**f**) Tegmina of the female individuals of *S. rostellocerca* showing the variation pattern of the tegmen among different individuals. (**g**–**l**) Tegmina of the types of *S. hengshanica*: (**g**) holotype male, (**h**,**i**) paratype males, (**j**–**l**) paratype females. Scale bar: 1 mm.

**Figure 4 insects-15-00526-f004:**
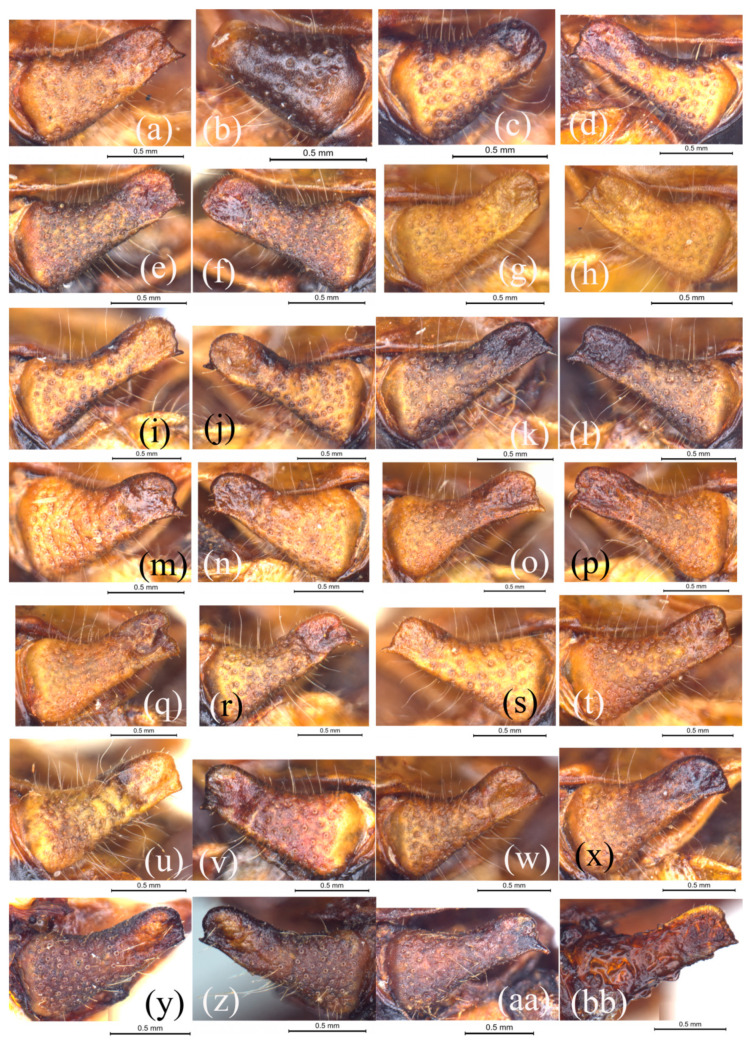
Variation of the cercus in the male *Sinopodisma rostellocerca* and *S. hengshanica*. (**a**–**l**) Cerci of six male individuals of *S. rostellocerca* showing the asymmetrical variation between the left and right cerci of the same individuals. (**m**–**x**) Cerci of the male individuals of *S. rostellocerca* showing the variation pattern of the cercus among different individuals. (**y**–**bb**) Cerci of the male types of *S. hengshanica*: (**y**,**z**) holotype male, (**aa**,**bb**) paratype males. Images (**a**–**p**,**y**,**z**) represent the left and right tegmina of the same individual, respectively.

**Figure 5 insects-15-00526-f005:**
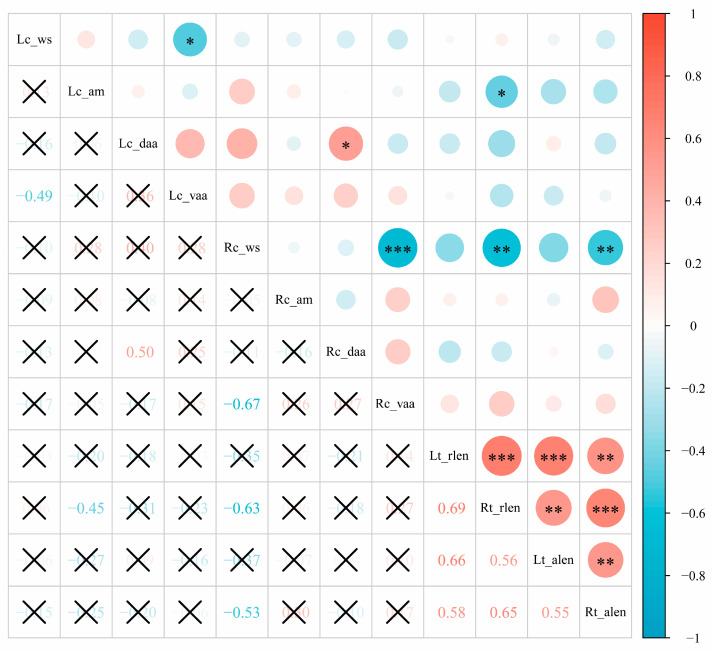
Correlation between tegmen and cercus of male *Sinopodisma restellocerca* and *S. hengshanica*. The asterisks in the figure indicate significantly correlated and the symbol “×” indicate no significant correlation.

**Figure 6 insects-15-00526-f006:**
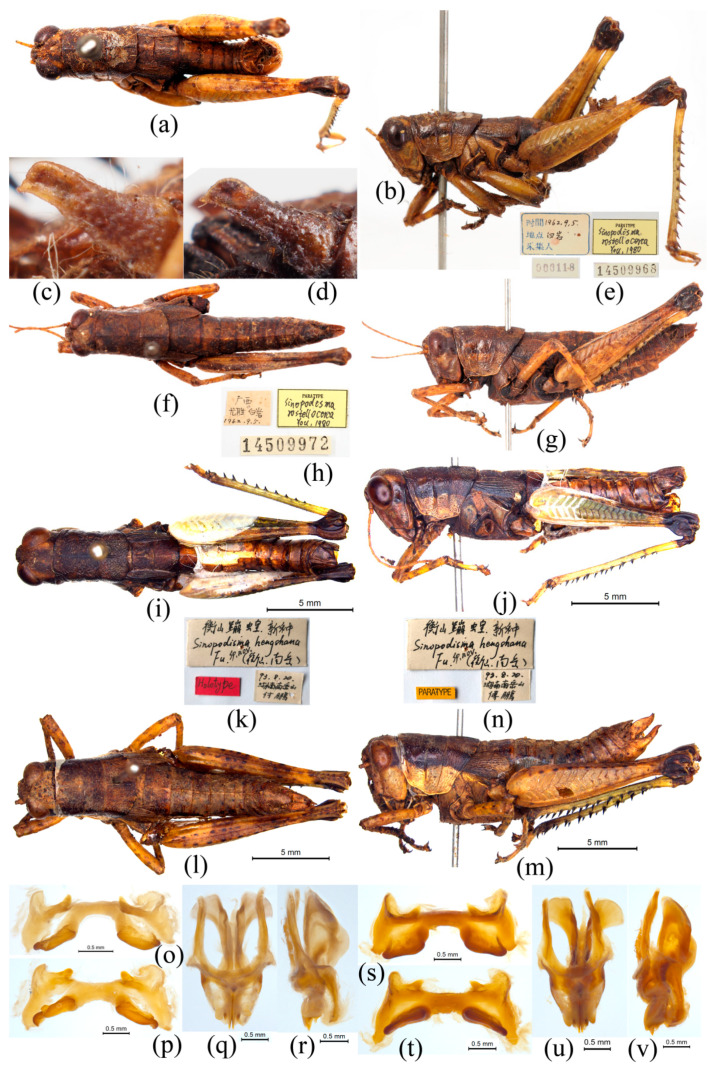
Habitus of types. (**a**–**h**) Types of *Sinopodisma rostellocerca*. (**a**,**b**) A paratype male in dorsal and lateral views. (**c**,**d**) Cerci of two paratype males in lateral views. (**e**) Labels of a paratype male; the Chinese in the label indicates the collecting date: 5 September 1962; the locality: Baiyan, Huaping National Nature Reserve, Longsheng County, Guangxi, China; collector: unknown. (**f**,**g**) A paratype female. (**h**) Labels of a paratype female, the Chinese in the label indicates the same information as in Figure 6e. (**i**–**n**) Types of S. *hengshanica*. (**i**,**j**) A holotype male in dorsal and lateral views. (**k**) Labels of a holotype male, the Chinese in the label indicates the Chinese common name of S. *hengshanica* the collecting date: 20 August 1993; the locality: Hengshan, Hunan Province, China; the collector: Peng Fu. (**l**,**m**) A paratype female in dorsal and lateral views. (**n**) Labels of a paratype female, the Chinese in the label indicates the same information as in Figure 6k. (**o**–**r**) Genitalia of *S. rostellocerca*. (**o**,**p**) Epiphallus in anterodorsal and dorsal views. (**q**,**r**) Phallic complex in dorsal and lateral views. (**s**–**v**) Genitalia of a paratype male of *S. hengshanica*. (**s**,**t**) Epiphallus in anterodorsal and dorsal views. (**u**,**v**) Phallic complex in dorsal and lateral views.

**Figure 7 insects-15-00526-f007:**
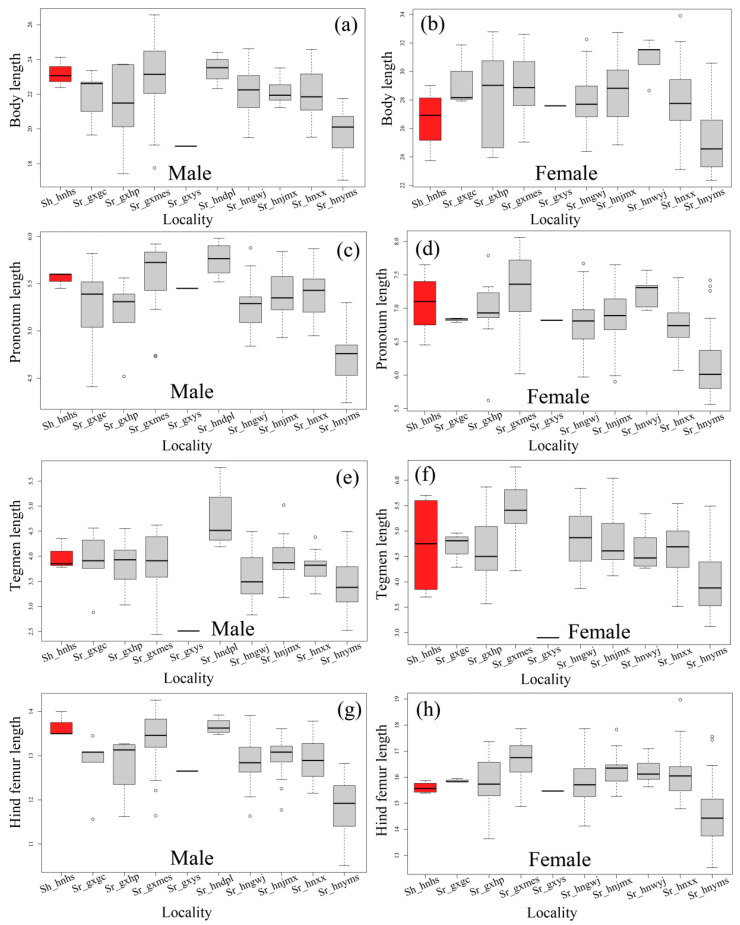
Box-and-whisker plots of four measurements of *Sinopodisma rostellocerca* and *S. hengshanica* generally used in the taxonomy of grasshoppers. (**a**,**b**) Body length. (**c**,**d**) Pronotum length. (**e**,**f**) Tegmen length. (**g**,**h**) Hind femur length. The red color indicates *S. hengshanica* and the grey color indicates *S. rostellocerca*. The localities represented by population identities as the labels of the X axis are as follows. Sr_hnxx: Xiaoxi, Yongshun, Hunan, China; Sr_hngwj: Gaowangjie, Guzhang, Huan, China; Sr_hnjmx: Jiemuxi, Yuanling, Hunan, China; Sr_hnyms: Yangmingshan, Shuangpai, Hunan, China; Sr_hndpl: Dupangling, Jiangyong, Hunan, China; Sr_gxmes: Mao’ershan, Xing’an, Guangxi, China; Sr_gxgc: Gongcheng, Guangxi, China; Sr_gxhp: Huaping, Longsheng, Guangxi, China; Sr_ptype: indicates the paratypes collected in Huping, Longsheng, Guangxi, China: Sr_gxys: Yaoshan, Guilin, Guangxi, China; Sh_hnhs: Hengshan, Hunan, China. Sr is the abbreviation of *S. rostellocerca* and Sh is that of *S. hengshanica*.

**Table 1 insects-15-00526-t001:** Number of specimens of *Sinopodisma* spp. examined in this study.

Species	Collecting Locality	Number of Individuals Examined		Category of Specimen
		Male	Female	
*S. rostellocerca*	Xiaoxi, Yongshun, Hunan, China	19	35	Common
*S. rostellocerca*	Gaowangjie, Guzhang, Hunan, China	33	73	Common
*S. rostellocerca*	Jiemuxi, Yuanling, Hunan, China	19	25	Common
*S. rostellocerca*	Yangmingshan, Shuangpai, Hunan, China	25	22	Common
*S. rostellocerca*	Dupangling, Jiangyong, Hunan, China	4	5	Common
*S. rostellocerca*	Maoershan, Xing’an, Guangxi, China	16	15	Common
*S. rostellocerca*	Gongcheng, Guangxi, China	5	3	Common
*S. rostellocerca*	Huaping, Longsheng, Guangxi, China	9 (5 + 4)	11 (10 + 1)	Common +Type
*S. rostellocerca*	Yaoshan, Guilin, Guangxi, China	1	1	Common
*S. hengshanica*	Hengshan, Hunan, China	3	4	Type
Sum		134	194	----

**Table 2 insects-15-00526-t002:** Coding matrix for correlation analysis between tegmen and cercus of male in *Sinopodisma restellocerca* and *S. hengshanica*.

Sample	Lc_ws	Lc_am	Lc_daa	Lc_vaa	Rc_ws	Rc_am	Rc_daa	Rc_vaa	Lt_rlen	Rt_rlen	Lt_alen	Rt_alen
Sample 1	1	1	2	5	1	3	1	4	7	7	4.5	4.1
Sample 2	1	1	1	4	1	5	1	4	7	8	4.2	4.7
Sample 3	1	1	1	2	1	1	1	2	8	8	4.8	4.2
Sample-ao4	1	1	1	3	1	1	1	3	6	6	3.9	4.6
Sample 4	1	2	1	5	1	2	2	5	6	5	3.9	3.5
Sample 5	1	2	1	4	3	2	1	2	1	2	3.1	3.5
Sample 6	1	2	1	4	1	1	1	4	5	5	3.7	3.6
Sample 7	1	2	1	4	1	2	1	4	7	7	3.6	4.0
Sample 8	1	2	1	5	1	2	1	5	5	5	3.8	3.8
Sample 9	4	2	1	2	1	2	1	4	6	5	4.2	3.7
Sample 10	1	2	1	4	1	2	1	4	5	5	4.3	3.7
Sample 11	1	3	1	4	1	3	1	4	8	3	4.2	4.2
Sample 12	1	4	1	4	1	1	1	5	6	5	3.8	3.7
Sample 13	1	4	1	3	1	2	1	5	5	5	3.85	3.8
Sample 14	1	4	2	3	1	2	2	4	3	3	3.8	3.7
Sample 15	2	4	1	2	1	1	1	3	8	8	4.8	4.7
Sample 16	1	3	1	5	1	6	1	5	7	7	4.2	4.6
Sample 17	5	3	1	2	1	2	1	3	5	6	3.5	3.5
Sample 18	1	1	1	2	1	1	1	5	6	7	4.3	3.45
Sample 19	1	4	2	5	6	2	1	1	5	1	3.6	2.9
S_hen_hm	1	3	1	2	1	6	1	5	5	5	3.65	3.9
S_hen_pm1	1	1	2	5	1	1	2	5	6	6	4.5	4.2
S_hen_pm2	1	2	1	4	1	2	1	4	8	8	3.7	4.0

Note: identifiers of the selected characters for correlation analysis are as follows: Lc, Rc, Lt, and Rt indicate the left cercus, right cercus, left tegmen, and right tegmen, respectively; ws, am, daa, vaa, rlen, and alen indicate whole shape of the apex of cercus, apical margin, dorso-apical angle, ventro-apical angle, relative length, and absolute length, respectively. The unit of the absolute length of the tegmen is in mm.

**Table 3 insects-15-00526-t003:** Measurement of *Sinopodisma rostellocerca* and *S. hengshanica* recorded in the literature.

Species	Body Length		Pronotum Length		Tegmen Length		Hind Femur Length	
	Male	Female	Male	Female	Male	Female	Male	Female
*S. rostellocerca* [3]	19.0–20.0	25.0–25.7	5.0	7.0–7.5	3.8	4.9	12.8	16.0
*S. rostellocerca* [25]	18.0–21.0	25.0–29.0	5.0–5.5	6.0–8.0	3.0–3.8	4.0–6.0	11.0–13.0	13.5–16.0
*S. hengshanica* [14]	22.7–23.0	31.0–31.5	5.5–5.7	7.2–7.3	4.0–4.1	3.3–3.5	13.5	17.5

Note: the unit of the measurement is in mm.

**Table 4 insects-15-00526-t004:** The mean and standard deviation of the measurement in different populations of *Sinopodisma rostellocerca* and *S. hengshanica*.

Pop_id	Body Length		Pronotum Length		Tegmen Length		Hind Femur Length	
	Male	Female	Male	Female	Male	Female	Male	Female
Sr_hnxx	22.02 ± 1.43	28.03 ± 2.21	5.38 ± 0.24	6.76 ± 0.29	3.77 ± 0.28	4.67 ± 0.49	12.92 ± 0.48	16.07 ± 0.82
Sr_hngwj	22.14 ± 1.38	27.85 ± 1.56	5.26 ± 0.23	6.76 ± 0.35	3.62 ± 0.46	4.85 ± 0.5	12.92 ± 0.49	15.83 ± 0.76
Sr_hnjmx	22.12 ± 0.62	28.6 ± 2.15	5.38 ± 0.24	6.92 ± 0.42	3.95 ± 0.4	4.82 ± 0.53	12.95 ± 0.44	16.26 ± 0.62
Sr_hnyms	19.71 ± 1.35	25.18 ± 2.47	4.7 ± 0.26	6.19 ± 0.57	3.44 ± 0.48	4.08 ± 0.67	11.85 ± 0.61	14.67 ± 1.27
Sr_hndpl	23.44 ± 0.86	30.88 ± 1.39	5.76 ± 0.19	7.24 ± 0.25	4.75 ± 0.70	4.65 ± 0.45	13.66 ± 0.19	16.26 ± 0.57
Sr_gxmes	22.96 ± 2.25	29.07 ± 2.16	5.57 ± 0.38	7.31 ± 0.52	3.85 ± 0.65	5.42 ± 0.56	13.35 ± 0.71	16.66 ± 0.82
Sr_gxgc	21.86 ± 1.51	29.32 ± 2.21	5.24 ± 0.54	6.83 ± 0.03	3.89 ± 0.65	4.69 ± 0.35	12.80 ± 0.73	15.87 ± 0.07
Sr_gxhp	21.29 ± 2.65	28.38 ± 3.26	5.17 ± 0.4	6.93 ± 0.56	3.83 ± 0.58	4.63 ± 0.65	12.72 ± 0.72	15.85 ± 1.06
Sr_ptype	18.75 ± 1.46	24.30	4.75 ± 0.33	6.60	3.38 ± 0.76	3.90	12.43 ± 0.77	16.10
Sr_gxys	19.01	27.59	5.45	6.82	2.51	2.90	12.65	15.47
Sh_hnhs	23.19 ± 0.87	26.66 ± 2.19	5.55 ± 0.09	7.08 ± 0.49	3.99 ± 0.31	4.72 ± 1.02	13.67 ± 0.29	15.6 ± 0.21
Collectivity	21.66 ± 1.87	27.87 ± 2.31	5.23± 0.41	6.79 ± 0.48	3.73 ± 0.55	4.74 ± 0.63	12.79 ± 0.74	15.86 ± 0.96

Note: The localities represented by population identities are as follows. Sr_hnxx: Xiaoxi, Yongshun, Hunan, China; Sr_hngwj: Gaowangjie, Guzhang, Huan, China; Sr_hnjmx: Jiemuxi, Yuanling, Hunan, China; Sr_hnyms: Yangmingshan, Shuangpai, Hunan, China; Sr_hndpl: Dupangling, Jiangyong, Hunan, China; Sr_gxmes: Mao’ershan, Xing’an, Guangxi, China; Sr_gxgc: Gongcheng, Guangxi, China; Sr_gxhp: Huaping, Longsheng, Guangxi, China; Sr_ptype: indicates the paratypes collected in Huping, Longsheng, Guangxi, China: Sr_gxys: Yaoshan, Guilin, Guangxi, China; Sh_hnhs: Hengshan, Hunan, China. Sr is the abbreviation of *S. rostellocerca* and Sh is that of *S. hengshanica*. The unit of the measurement is in mm.

**Table 5 insects-15-00526-t005:** The extremum of the measurement of *Sinopodisma rostellocerca* and *S. hengshanica*.

	Body Length (Min–Max)		Pronotum Length (Min–Max)		Tegmen Length (Min–Max)		Hind Femur Length (Min–Max)	
	Male	Female	Male	Female	Male	Female	Male	Female
Sr_hnxx	19.53–24.58	23.12–33.91	4.95–5.87	6.07–7.46	3.25–4.38	3.52–5.54	12.15–13.78	14.79–18.97
Sr_hngwj	19.49–24.61	24.39–32.25	4.84–5.88	5.97–7.67	2.83–4.49	3.87–5.84	11.63–13.91	14.13–17.86
Sr_hnjmx	21.24–23.51	24.85–32.73	4.93–5.84	5.90–7.65	3.18–5.02	4.12–6.04	11.77–13.61	15.27–17.83
Sr_hnyms	17.05–21.75	22.36–30.58	4.24–5.3	5.56–7.42	2.52–4.49	3.12–5.49	10.51–12.82	12.53–17.56
Sr_hndpl	22.32–24.41	28.66–32.20	5.52–5.98	6.97–7.57	4.19–5.77	4.27–5.34	13.48–13.92	15.63–17.1
Sr_gxmes	17.76–26.56	25.04–32.61	4.73–5.92	6.02–8.06	2.44–4.62	4.22–6.26	11.64–14.26	14.88–17.86
Sr_gxgc	19.65–23.36	27.92–31.87	4.41–5.82	6.79–6.85	2.88–4.56	4.29–4.96	11.56–13.45	15.82–15.95
Sr_gxhp	17.00–23.72	23.96–32.79	4.40–5.56	5.62–7.79	2.90–4.55	3.57–5.87	11.60–13.40	13.64–17.36
Sr_ptype	17.00–20	24.30	4.40–5.20	6.60	2.90–4.50	3.90	11.60–13.40	16.10
Sr_gxys	19.01	27.59	5.45	6.82	2.51	2.9	12.65	15.47
Sh_hnhs	22.39–24.12	23.75–29.02	5.45–5.60	6.45–7.65	3.78–4.35	3.70–5.7	13.50–14.00	15.39–15.87
Collectivity	17.00–26.56	22.36–33.91	4.24–5.98	5.56–8.06	2.44–5.77	2.90–6.26	10.51–14.26	12.53–18.97

Note: The localities represented by population identities are as follows. Sr_hnxx: Xiaoxi, Yongshun, Hunan, China; Sr_hngwj: Gaowangjie, Guzhang, Huan, China; Sr_hnjmx: Jiemuxi, Yuanling, Hunan, China; Sr_hnyms: Yangmingshan, Shuangpai, Hunan, China; Sr_hndpl: Dupangling, Jiangyong, Hunan, China; Sr_gxmes: Mao’ershan, Xing’an, Guangxi, China; Sr_gxgc: Gongcheng, Guangxi, China; Sr_gxhp: Huaping, Longsheng, Guangxi, China; Sr_ptype: indicates the paratypes collected in Huping, Longsheng, Guangxi, China: Sr_gxys: Yaoshan, Guilin, Guangxi, China; Sh_hnhs: Hengshan, Hunan, China. Sr is the abbreviation of *S. rostellocerca* and Sh is that of *S. hengshanica*. The unit of the measurement is in mm.

## Data Availability

Original data of measurement available from one of the corresponding authors, Jianhua Huang.

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
