# Peer review of "Variation of the Tegmen and Cercus in Sinopodisma rostellocerca (Orthoptera: Acrididae: Melanoplinae) with Proposal of a New Synonym"

_insects, 2024, doi:10.3390/insects15070526_

Round 1
Reviewer 1 Report
Comments and Suggestions for Authors
The contents of the manuscript are okay but several corrections of wording and spelling are required for correct English grammar.
Comments on the Quality of English Language(1) remark on title:
the title could be shortened to:
Variation of Tegmen and Cercus in Sinopodisma rostellocerca (Orthoptera: Acrididae: Melonoplinae) with Proposal of a New Synonym
(2) general remarks and corrections of words appearing many times in the manuscript:
variation does not take a plural form (variations), it is always variation
Material, material does not take a plural form (materials) it is always material
corner, corners: better use the word angle, angles
(3) proposed corrections of text with line numbers:
24 shewed SHOULD BE: showed
30 was herein SHOULD BE: is herein
36-37 all the remaining SHOULD BE: all remaining
39 in northern region SHOULD BE: in the northern region
40 and southwestern region SHOULD BE: and the southwestern region
49-50 our wild investigations SHOULD BE: our investigations in the field
52-53 intrapopulation individuals SHOULD BE: individuals of the same population
56 we considered herein S. hengshanica a new junior synonym SHOULD BE: we consider S. hengshanica to be a new junior synonym
58 based on the specimens SHOULD BE: based on specimens
59 morphological terminology SHOULD BE: terminology for morphology
63-64 the stacking images were combined SHOULD BE: the images were stacked
75 variations always exist SHOULD BE: variation exist
98-99 the majority of individuals are SHOULD BE: the majority of individuals is
101 reach only the half or the two thirds SHOULD BE: reach only half or two thirds
113 individuals with the apical margins SHOULD BE: individuals having the apical margins
114 and one with SHOULD BE: and one having
135 kinds of situations SHOULD BE: kinds of situation
139 can be involved in the range SHOULD BE: can also occur
142 variation in male cercus SHOULD BE: variation of the male cercus
144 looks completely a different appearance SHOULD BE: shows a completely different appearance
147 the left and right ones SHOULD BE: the left and right cerci
149 the opposite ones SHOULD BE: the opposite cerci
151 left and right ones SHOULD BE: left and right cerci
183 completely agree with SHOULD BE: completely agrees with
185 the left one SHOULD BE: the left cercus
186 the right one SHOULD BE: the right cercus
187 is angulate in the left SHOULD BE: is angulate at the left
187 but denticulate in the right SHOULD BE: but denticulate at the right cercus
203 reclinate in profile view SHOULD BE: reclined in profile
(Remark: the word reclinate does not exist, do you mean reclined ?)
206 slender SHOULD BE: narrow
212 slightly straight SHOULD BE: nearly straight
216-217 with the shape of apex usually varing among species SHOULD BE: the shape of the apex usually varies within species
234 below the median oculus SHOULD BE: below median oculus
234-235 with fastigium SHOULD BE: with fastigium verticis
238 beyond the posterior SHOULD BE: beyond posterior
240 punctated SHOULD BE: punctate
240 circular arced SHOULD BE: circularly arced
241 notch in the middle SHOULD BE: notch in middle
242 with the posterior one SHOULD BE: with the posterior sulcus
246 Both tegmina and SHOULD BE: Both, tegmina and
256 covered completely SHOULD BE: completely covered
260 punctated SHOULD BE: punctate
261 furcula at the middle of the posterior margin SHOULD BE: furcula in the middle of posterior margin
270 except the SHOULD BE: except for the
275 near the base SHOULD BE: near base
290 but not the apex SHOULD BE: but not at the apex
290-291 and the main difference is the apex of cercus SHOULD BE: the main difference being the apex of cercus being
301 within a species and defined as at least partly independent SHOULD BE: within a species and is defined as being at least partly independent
307 Despite the prevalence SHOULD BE: Despite of the prevalence
322 male genitalic shape SHOULD BE: the shape of male genitals
334 sometimeeven SHOULD BE: sometimes even
335-336 do not certainly represent independent species SHOULD BE: do not represent independent species with certainty
342 as phylogeny reconstruction SHOULD BE: as is phylogenetic reconstruction
350 quantitative analysis such as morphometric SHOULD BE: quantitative analysis using morphometric
351 help distinguish interspecific and intraspecific variations SHOULD BE: help to distinguish between interspecific and intraspecific variation
358 molecular material SHOULD BE: molecular data
359-360 data is expected SHOULD BE: data are expected
Author Response
- The title of the manuscript has been shortened according to the remarkof the reviewer 1.
- The word “variations”in an incorrect plural form all over the main text has been replaced with “variation”in a correct singular form.
- The word “materials”in an incorrect plural form all over the main text has been replaced with “material”in a correct singular form.
- The words “corner”and “corners”have been replaced with “angle” and “angles” respectively according to the suggestion of the reviewer 1.
- All other corrections of text proposed by the reviewer 1 have been fully accepted.
We fervently acknowledge the detailed suggestions by the reviewer for us to improve the writing of the manuscript.
Reviewer 2 Report
Comments and Suggestions for Authors
The author's attempt to study the intraspecific variation of wing shape and cerci is a highly meaningful endeavor. It plays a positive role in understanding the patterns of intraspecific and interspecific morphological variation, improving classification methods, and diagnostic features. However, there are still some important issues that need to be addressed.
Firstly, the presentation of the three figures illustrating wing shape variation needs improvement. I will suggest to use a single scale for the left and right wings of the same individual, allowing for a clear display of size differences within a single figure. The figure captions should also be more detailed and clear, indicating which wings belong to the same specimen.
Secondly, when discussing inter-individual variation of a species, sample size is a crucial parameter. Without an adequate sample size, it is difficult to measure all the differences and variations accurately. The author did not specify the number of samples included in the statistical analysis, which should be clearly stated in the Materials and Methods section. For extant species, a large sample size is necessary, and it is suggested to include at least 5 or 10 specimens of both males and females for meaningful analysis.
Furthermore, why did the authors only consider wing outline but without wing venation? Both fossil and extant Orthoptera have shown that wing venation features have certain taxonomic value. The author should attempt to compare intra/inter-specific differences of wing venation. It is highly recommended for the author to utilize geometric morphometrics to study the variation patterns of both wing outline and venation. When discussing variation, quantitative analysis is necessary, and combining it with comparative morphology may provide better answers to the questions raised by the author. Additionally, it is suggested for the author to include a linedrawing for illustrating wing venation characteristics.
The discussion, particularly section 4.1, appears lengthy and ineffective, resembling more of a review rather than a discussion based on the results of this study. It is recommended to focus the discussion on the viewpoints from the perspectives of interspecific differences and intraspecific variation.
Other issues: In Figure 5, it is advised to align all specimens in the same direction, such as having the head on the left and the abdomen tip on the right.
Line 299: "[?]" Please provide the missing reference.
It seems that the author's literature review is not comprehensive enough. Studies such as Wen et al., 2015, which investigated wing morphological difference in grasshoppers at different taxonomic levels by geometrics, is very related to your study, and should be introduced and compared in this study.
Best wishes
Comments on the Quality of English Language
I am not qualified to assess the quality of English in this paper
Author Response
- For the first suggestion to use a single scale for the left and right wings of the same individual, it is indeed a good recommendation. We justified the size of the images to ensure the scale of each image being identical and then used a single scale for all images on the same plate.
- The figure captions havebeenrevised being more detailed to indicate which tegmina belong to the same individual.
- The sample sizehas been clearly complementedin the Material and Methods section.
- For the application of the wing venation, the current status is that the wingvenation has never been used in Catantopidae (sensu lato) with short lateral tegmen for taxonomic purpose due to the reduction of viens.
As for the outline of the tegmen, it indeed has significant value for separating good species. However, we believe that the reliability of our results in this manuscript will not be affect by not using geometric morphometrics, because only a single species with its newly proposed junior synonym was dealt with in the manuscript and the distinct intraspecific variation has been fully illustrated. Of course, we will accept the recommendation of the reviewer in the future to utilize geometric morphometrics in the more comprehensive studies to explore the significance of the shape of tegmen in distinguishing good species just like we do the quantitative analysis on the variation in the measurement of four characters generally used in grasshoppers and the correlation analysis between the tegmen and cercus in male.
- The discussionsection has been revised carefully according to the comment of the reviewer.
- In Figure 5, all specimens have been alignedin the same direction, having the head on the left and the abdomen tip on the right.
- In line 299, the missing referencehas been added.
- Some more references on discussing intraspecific variation of grasshoppers have been complemented to the manuscript, including the one “Wen et al., 2015” proposed by the reviewer.
Reviewer 3 Report
Comments and Suggestions for Authors
The manuscript “Morphological Variations of Tegmen and Cercus in Sinopodisma rostellocerca (Orthoptera: Acrididae: Melanoplinae) with Proposal of a New Synonym” documents variation in tegmental and cercal morphology between two species, Sinopodisma rostellocerca and S. hengshanica, and subsumes diagnostic features among the structural variation, as the main conclusion of the study proposing a single species S. rostellocerca.
The study addresses the taxonomic discrimination of acridid species of Sinopodisma, and contributes to the species identification and taxonomy of this group. As such, it is relevant work in the context of insect biodiversity and insect morphology. In the present form, however, the manuscript has certainly shortcomings in the data analysis and presentation / discussion, which need to be addressed. As a main point, the data presented here are descriptions of morphological types, while quantitative parameters like length are only mentioned but not measured. This makes the manuscript in the present form appear as a somewhat preliminary study. The question of the study is presented with as a rather specialised taxonomic problem in the introduction. Some broader explanation would be useful why these species are particularly relevant for identification, as the present manuscript seems to address a specialised taxonomic readership. Therefore, both the data analysis and the presentation should be improved.
General aspects that need to be improved:
Introduction: while the abstract points at an apparent neglect of variation by evolutionary biologists, this is not taken up in the introduction. Rather, the manuscript presents a specific taxonomic problem of species identification among Sinopodisma (l. 43ff). This is not a very homogenous presentation, and since the taxonomy is the main focus, the statement in the abstract on neglect of variation by systematists and evolutionary biologists (l. 19 – 20) should be explained better or toned down – after all, variation is a crucial concept for evolutionary analysis. The introduction should focus directly on the species were the species status is uncertain and introduce the diagnostic characters, and then give further details like the overall species number etc. What about the structure of the insect genitalia, have they been analysed for potential morphological differences?
Several minor explanations are missing in the introduction, e.g., the terminology on the cerci should be established in the introduction (“rostriform” is only used in the methods (l. 143), “frustum-cone-like” is not explained (l. 146), and “denticulate” is relating to a main morphological difference (l. 291) but not used in introduction or in the analysis of cercal shapes in section 3.2) – this should all be outlined when the hypothesis on different morphological traits is presented in the introduction, and included in the description of results. Further, “differences between these three species are extremely minute” (l. 47) but there is no information on these differences and how they relate to the present study, the “so-called distinguishing characters” (l. 54) should be explained and referenced. There is information on distinguishing cercal form only later in the results (l. 291), and all of this should be coherently presented in the presentation to explain why wings and cerci are analysed here. Some more information could also be presented on the taxon investigated, e.g., that Melanoplinae commonly have short wings or are apterous (Storozhenko 1993 Articulata 8: 1–22; Rivera García 2006 Acta Zoológica Mexicana 22: 131–149; Lehmann 2010 J Comp Physiol A 196: 807–816).
Other pieces of text appears scattered, consider for example moving the text lines 73 – 76 and lines 318 – 322 to the introduction as some more general information.
Materials and methods: The manuscript lacks information how many individuals were photographed / inspected for each species and sex – this should be definitely included in the materials section, or in the results.
The data presented here are rather qualitative in nature, and this raises the question if some more quantitative data can be derived from this that would allow for a statistical analysis – as an example, see the statement in the results on length differences of tegmen (l. 100). Consider measuring the tegmen and cercus length or diameter of cercus base and / or tip, for example, or find some relevant parameters for the shape.
Results: this section should consider quantifying the variation as noted above. It also states for tegmen that most individuals were in fact symmetric. Also, revise the figure panels to identical sizes of individual figures.
Fig. 5 could be moved up to give an overview of the insects’ habitus and relevant features.
Discussion: it presents some general considerations and has only few direct correlations to the new results. The main finding should be stated shortly at the beginning of the discussion, and the discussion of the new findings could then combine the parts of lines 330 – 337 and lines 346 – 349. The remaining discussion of species status and methods should be condensed to specific questions regarding the Sinopodisma species.
Very interesting would be the behaviour of insects – have mating attempts been observed in the field between S. rostellocerca and S. hengshanica, or could this be tested in the laboratory in choice experiments?
Minor:
L 17 …was herein considered / identified as a new junior synonym…
L 24 The results showed…
L 41 several species – how many? Are they generally closely related and overall similar in morphology?
L 53 The similar variation – rephrase, do you mean: “similar variation”? has this been quantified?
L 101, 102 Figure 1c, d, i; Figure 1e…
L 142 Variation in the male cerci is more extensive than in [female cerci or in tegmen etc]
L 150 as this is later relevant for discussion, is any shape liking to the denticulate form mentioned only later?
L 222 include reference to Ref. [2]
L 267 more robust
L 288 reference is missing in the sentence.
Author Response
We are extremely grateful to the pertinent comment by the reviewer on the manuscript. Now we answer, as a whole but not one by one, the reviewer’s comment. According to the suggestion by this reviewer in combination with those by other reviewer’s, we have made a major revision to the manuscript. We have moved some text on intraspecific variation in discussion section ahead to introduction section to explain in detail the status of the neglect of variation by systematists and evolutionary biologists. The number of material examined, carefully compared and photographed has been provided in detail in Material and Methods section, the quantitative analysis of the measurement generally used in grasshoppers such as body length and the correlation analysis between male cercus and tegmen have been implemented according to the suggestion by the reviewer and some other reviewers, with the detailed approaches introduced in Material and Methods section. We certainly used the terminology such as “rostriform”, “frustum-cone-like”, “denticulate” to describe our result but not only in the methods section. For the statement “differences between these three species are extremely minute”, we have pointed out in the brackets where the detailed information on this minute difference has been provided and we think this is clear in meaning. After that, we further pointed out that “distinct intraspecific variation in the shape of tegmen and male cercus were observed frequently in S. rostellocerca during our investigations in the field, making the boundaries among these species more ambiguous”, and this is how they relate to the present study and why tegmen and cercus are analysed here. We have quantified the shape of male cercus and the relative length of tegmen to generate a coding matrix and carried out based on this matrix a correlation analysis between these two characters. We also made a statistical analysis of the four measurement generally used in grasshoppers, including the mean, standard deviation, extremum value and variance analysis of each single character between population pairs. We have revised the discussion section according to the suggestion by this and some other reviewers.
In a word, we have tried our best to revise the manuscript and we think the present status of the manuscript has looked much better than the first version submitted. Now we think our revised manuscript is ready for resubmitted to Insect.
As for the question the reviewer asked in the last paragraph, the present circumstance is that no additional material of S. hengshanicahas has been obtained again except the types despite we go to the type locality, Hengshan, several times during the past years. We can’t be able to provide the behavior data as the reviewer asked. We are extremely sorry for this. However, we think this will not affect the reliability of our taxonomic decision because we have provided enough evidence to support our decision and the behaviour data is not always a necessary evidence for species discrimination.
All other minor suggestion have been accepted and corresponding revision has been made.
Thank you very much for you comment!
Reviewer 4 Report
Comments and Suggestions for Authors
Intraspecific variation is ubiquitous from individual traits to population level and plays an important role in a variety of fields. In this paper, the intraspecific variations of tegmen and cercus in Sinopodisma rostellocerca were examined, and S. hengshanica was herein treated as a new junior synonym of S. rostellocerca. It is worthy to publish this contribution. However before it is formally accepted, there are some issues needed authors to make it clear.
In figures, Cloud the images be standardized to a single scale to facilitate comparison of morphological sizes? It may be possible to omit the 1mm scale within the image, with the scale noted at the end of the legend as "scale: 1mm".
Should this article specify the exact number of specimens examined for each species and the proportion of variation observed? Providing numerical data can offer a more objective illustration of the issue.
Lines 100-115 mention lengths but do not provide actual numerical values. This information needs to be supplemented.
In lines 72, 129, 141, 182, 194, and 195, the scientific names in the headings should be italicized.
In Figure 4, are the cerci corresponding to the wings mentioned earlier or to other individuals? If they pertain to other individuals, why aren't the cerci of the individuals with morphological variation in the wings jointly analyzed? Is there a significant difference in their cerci morphology?
This article only discusses morphological variations in wings and cerci, leading to the reclassification of S. hengshanica as a junior synonym of S. rostellocerca. It does not demonstrate whether there are differences in other valid classification features, such as body size. Previous literature indicates that the two species' morphometric data do not overlap; compared to S. rostellocerca, S. hengshanica tends to be slightly larger. Additionally, are there differences in epiphallus and phallic complex between the two species? In grasshopper taxonomy, male reproductive organs are significant classification features, and previous literature descriptions indicate clear differences in epiphallus and phallic complex of these two species.
In line 228, should "et al" be italicized? This is inconsistent throughout the article. Please carefully review and rewrite according to the MDPI template format.
Lines 231-249: Please supplement the article with detailed images and refer to them in the description, including images of the head in frontal view, antennae, dorsal and ventral views of the pronotum, anal plate and cerci, as well as some aspects of the females. This will provide readers with a clearer understanding of the descriptions.
In line 241, please ensure consistency in singular and plural forms for "lateral carina" and "median carina".
Lines 245-246: What is the proportion of mesosternal lobes with interspace?
Should the sentence in line 267, " similar to male", subsequent descriptions diverge from male characteristics in some features, be placed at the end of the paragraph?
Regarding lines 272-275, the measurements for S. hengshanica are 22.7-23mm for males and 31-31.5mm for females, which are evidently larger than the measurements for S. rostellocerca (including other measurements). If S. hengshanica is indeed a junior synonym of S. rostellocerca, should these measurements be included and verified for accuracy?
What does [?] represent in line 288?
The distribution areas of S. rostellocerca and S. hengshanica are vastly separated, with no overlap. The Nanling Mountains in central China act as a barrier, with many other species of grasshoppers, inhabiting the area. How can such distribution patterns be explained?
Please review the references, including numbering, journal name abbreviations, and italicization of scientific names.
Are there data from non-type specimens of S. hengshanica available to support the validity of this junior synonym? Do these specimens primarily exhibit characteristics of S. hengshanica or S. rostellocerca?
Author Response
- For the first suggestion to use a single scale for all images in the same plate, it is indeed a good recommendation. The images in the same plate have be standardized to a single scale according to the suggestion of the reviewer.We justified the size of the images to ensure the scale of each image being identical and then used a single scale for all images on the same plate.
- 2. The sample sizehas been clearly complementedin the Material and Methods section, and the proportion of variation has been provided in the results.
In addition, the measurement of body, pronotum, tegmen and hind femur have been updated and the mean, standard deviation and extremum of the measurements of the species and those for different populations have been supplemented, respectively.
- The scientific names in the headings have been resetaccording to the format of the journal.
- The cerci in Figure 4 were photoed from the same series of individuals for comparison of tegmen shape. The correlation between cercus and tegmen has been analyzed using quantitative methods and the correlation hot plot shows that there is nearly no correlation between the variation of cercus and tegmen.
- The reviewer stated that “This article only discusses morphological variations in wings and cerci, leading to the reclassification ofS. hengshanicaas a junior synonym of S. rostellocerca. it does not demonstrate whether there are differences in other valid classification features, such as body size, epiphallus and phallic complex”. According to the original description (Fu, 1998), the main character to distinguish S. hengshanica from its closest relatives, S. rostellocerca and S. spinocerca is the apical shape of male cercus. We have expatiated in the “remark” part in section 3.4.2 that there is no other useful character for separating these two species. We have also supplemented the the quantitative variation in the measurement of four characters generally used in grasshoppers using data from more than 300 individuals, implemented correlation analysis between cercus and tegmen and dissected the male genitalia of both species, and the result shows that there is also no significant difference between the two species in the usually used measurement and male genitalia. The measurement of S. rostellocerca.varied along with the increase of material examined in historical and this study and there is in fact no significant difference in body size between S. rostellocerca and S. hengshanica as showed in the result of quantitative analysis. As for male genitalia, in fact, the original description of S. rostellocerca didn’t provide description and illustration of male genitalia, and the difference in male genitalia between S. rostellocerca and S. hengshanica didn’t be mentioned when S. hengshanica was described by Fu (1998). As for the difference in male genitalia between S. hengshanica and S. spinocerca mentioned by Fu (1998), it does not affect the recognition and taxonomic decision of S. rostellocerca and S. hengshanica.
- Acording to the format of Insects, “et al”need not be italicized, and all have been revised consistentthroughout the manuscript.
- Lines231-249: while the reviewer’s recommendation is very nice, we do not supplement those images suggested by the reviewer at the moment, because the structures such as head in frontal view, antennae, dorsal and ventral views of the pronotumand anal plate are not the distinguishing characters between S. rostellocerca and S. hengshanica and the addition of these images will increase the number of plates and the length of the manuscript but we think this is not necessary at the moment. Of course we will accept the suggestion in future when necessary just like we have accepted most of suggestion by the reviewer during the course of the revision of this manuscript. We thank the reviewer very much for his nice recommendation!
- In line 241,we ensure that both "lateral carina" and "median carina"should be in singular form because for negative description “absent” the noun “carina” should be in singular form according to English grammar even though this noun should be in plural form when it is present. When the lateral carina is absent, how could it be in plural form since no one is present?
- the proportion of length and width of interspacebetween mesosternal lobes is measured and added to the manuscript. In line 267, the sentence " similar to male"has been placed at the end of the paragraph according to the comment of the reviewer.
- For the measurement of S. rostellocercaand S. hengshanica, this is our carelessness. We forgot to update the measurement at this part. According to the suggestion by the reviewer, we remeasured more than 300 individuals and compared the data at population level and found no diference between this two species. The measurement of S. rostellocercanow has been updated in the revised version of the manuscript.
- the missing reference in line 288 has been added.
- While the record of S. hengshanicanow is limited to Hengshan, the middle of Hunan province only, S. rostellocercahas been found distributed not only in the area south to Nanling Mountain, but also in the area north to Nanling Mountain such as Huaihua, Tongdao, Suining, Chengbu, Wugang, Shuangpai and Dupangling (Jiangyong couties) of southwest Hunan, and so on, and expanded to Taoyuan, Yuanling, Guzhang and Yongshun counties in northwest Hunan. Therefore, there is in fact no geographical barrier between the distribution areas of S. rostellocerca and S. hengshanica.
- the reference has been examined carefully according to the suggestion of the reviewer.
- We went to Hengshan several times and try to collect more material for both morphological and molecular study. However, no specimen was founded during the past several years. Therefore, we have no data from non-type specimens ofS. hengshanicaavailable to support the validity of this junior synonym. But we think this will not affect the validity of our taxonomic decision on the relationship between S. hengshanica and S. rostellocerca since so many material from so many localities have been examined and compared using traditional and quantitative approaches. Thank you for the comment of the reviewer!
Round 2
Reviewer 2 Report
Comments and Suggestions for Authors
Dear colleagues:
thank you for your reply, it is much better now, I have no more revision. congratulations.
Best wishes
JJ
Author Response
We checked the comment from the reviewer, and he/she stated that "it is much better now, I have no more revision". So we have no reply for any revision.
Reviewer 3 Report
Comments and Suggestions for Authors
“Morphological Variations of Tegmen and Cercus in Sinopodisma rostellocerca (Orthoptera: Acrididae: Melanoplinae) with Proposal of a New Synonym”:
The revised manuscript is now carefully edited and thoroughly expanded with further analysis and overall a clearly improved presentation of the data and of the discussion. However, the author response “as a whole” makes it really difficult to track specific revisions within the manuscript. For example, the on individual points, suggested references on Melanoplinae are not included, and some sections of text suggested to be moved to the introduction as more general information is still placed in results (l. 140-144) and discussion (l. 550 – 552). Consider at least the first section of those to be placed in a different section, because it is not a unique result of the present study.
The authors have added figures, explain now data in more detail, and outline the topic further in the introduction and discussion. Especially important was the added information on the number of specimen analysed and measured, as it is now stated in the manuscript.
In the current form, some points should be addressed in the text and figures, please see below:
l. 42 “beak-like” – this terminology is only used in introductory paragraph. Consider to include one section were all terms are introduced in one place, and linked to the relevant species.
l. 73 “in great number” – specify here, or rephrase otherwise.
l. 481 more than three hundreds individuals of (delete first „of“)
Figures: Please re-work some of the figures:
In the Figures 2, 3 and 4, individual photographs should be aligned, especially
Fig. 7: the labelling is far too small, please adjust font size. Also include in the figure legend what the red box shows for quick first orientation to readers.
Author Response
- The reviewer suggested us in the review of round 1 to include some more references on Melanoplinae to provide more informationon the taxon investigated. We are sorry for our oversight in the first round of revision due to the huge content to be revised and the rushing time for a majore revision. Now we add to the manuscript the reference Storozhenko (2019) suggested by the reviewer. Please see reference [5] and the text at the end of the first paragraph in introduction section.
-
We have moved the text at the beginning of the section “3.1.1.”, i.e. “As a group of bilaterally symmetrical organisms, insects have many binate internal and external organs which are usually equivalent to (or mirror images of) each other in shape and structures. However, intraspecific variation exists not only among different individuals but also between the different sides of the same individual sometimes, resulting in an asymmetry between the left and right components of the binate organ. ”, to the beginning of the last paragraph in introduction section.
-
While sometimes the use of synonyms is helpful in enriching the expression of the language, especially in common statement (sentence), it is really better to use special terminology in normative morphological description. Therefore, according to the suggestion by the reviewer, we replaced the word “beak-like” in introduction section with the more special word “rostriform” which is used later in result section.
-
We have added the paraphrase for the special terminology for describing the shape of cercus into Material and Methods section.
-
We have no way to specify the statement “...a large number of ...” and “...a great number of...” between the previous lines 71 and 74 because we directly cited the original sentence from the reference Cuezzo et al. (2020), which runs thus.
-
the word “of” in the sentence “...more than three hundreds of ... ” in line 481 has been deleted although our expression is also available.
-
We think there is no need to align the individual images in Figures 2, 3 and 4, because they were aligned originally but two of the reviewers in the review of round 1 requested us to use a single scale for the whole plate, so we justified the size of the images in the same plate to ensure the scale of each image being identical and then used a single scale for all images in the same plate. This is just a difference in academic point of view by different scholars.
-
The labeling in Figure 7 have been enlarge to an appropriate size. The box color representing different species has been explained in the capture.